# COOPERATIVE MINIBATCHING IN GRAPH NEURAL NETWORKS

## ABSTRACT

Significant computational resources are required to train Graph Neural Networks (GNNs) at a large scale, and the process is highly data-intensive. One of the most effective ways to reduce resource requirements is minibatch training coupled with graph sampling. GNNs have the unique property that items in a minibatch have overlapping data. However, the commonly implemented Independent Minibatching approach assigns each Processing Element (PE) its own minibatch to process, leading to duplicated computations and input data access across PEs. This amplifies the Neighborhood Explosion Phenomenon (NEP), which is the main bottleneck limiting scaling. To reduce the effects of NEP in the multi-PE setting, we propose a new approach called Cooperative Minibatching. Our approach capitalizes on the fact that the size of the sampled subgraph is a concave function of the batch size, leading to significant reductions in the amount of work per seed vertex as batch sizes increase. Hence, it is favorable for processors equipped with a fast interconnect to work on a large minibatch together as a single larger processor, instead of working on separate smaller minibatches, even though global batch size is identical. We also show how to take advantage of the same phenomenon in serial execution by generating dependent consecutive minibatches. Our experimental evaluations show up to 4x bandwidth savings for fetching vertex embeddings, by simply increasing this dependency without harming model convergence. Combining our proposed approaches, we achieve up to 64% speedup over Independent Minibatching on single-node multi-GPU systems.

## 1 INTRODUCTION

Graph Neural Networks (GNNs) have become de facto deep learning models for unstructured data, achieving state-of-the-art results on various application domains involving graph data such as recommendation systems (Wu et al., 2020; Ying et al., 2018), fraud detection (Liu et al., 2022; Patel et al., 2022), identity resolution (Xu et al., 2019), and traffic forecasting (Jiang & Luo, 2022). However, as the usage of technology continues to increase, the amount of data generated by these applications is growing exponentially, resulting in large and complex graphs that are infeasible or too time-consuming to train on a single processing element (Ying et al., 2018; Zhu et al., 2019). For example, some social media graphs are reaching billions of vertices and trillions of interactions (Ching et al., 2015). Efficient distributed training of GNNs is essential for extracting value from large-scale unstructured data that exceeds the cost of storing and processing such data.

Due to the popularity of Deep Neural Networks (DNNs) and the need to support larger models and datasets, a great deal of research has focused on increasing the scale and efficiency of distributed DNN training. Techniques such as data parallelism (Ginsburg et al., 2017; Goyal et al., 2018), pipelining (Narayanan et al., 2019), and intra-layer parallelism (Dean et al., 2012) have been employed. Following the success of traditional distributed DNN training, the same techniques have also been adapted to GNN training, such as data parallelism (Gandhi & Iyer, 2021; Lin et al., 2020; Zheng et al., 2021; Zhu et al., 2019) and intra-layer parallelism (Tripathy et al., 2020).

The parallelization techniques mentioned earlier are used to scale both full-batch and minibatch training in a distributed setting. Minibatch training (Bertsekas, 1994) is the go-to method to train DNN models as it outperforms full-batch training in terms of convergence (Allen-Zhu & Hazan, 2016; Li et al., 2014; Keskar et al., 2016; Wilson & Martinez, 2003), and more recently has been shown

to also offer the same benefit for GNNs (Zheng et al., 2021). In the distributed setting, minibatch training for DNNs using data parallelism is straightforward. The training samples are partitioned across the Processing Elements (PE) and they compute the forward/backward operations on their minibatches. The only communication required is an all-reduce operation for the gradients.

Unfortunately, minibatch training a GNN model is more challenging than a usual DNN model. GNNs turn a given graph encoding relationships into computational dependencies. Thus in an $L$-layer GNN model, each minibatch computation has a different structure as it is performed on the $L$-hop neighborhood of the minibatch vertices. Real-world graphs usually are power law graphs (Artico et al., 2020) with small diameters, thus it is a challenge to train deep GNN models as the $L$-hop neighborhood grows exponentially w.r.t. $L$, reaching almost the whole graph within a few hops.

Very large GNN datasets necessitate storing the graph and node embeddings on slower storage mediums. To enable GNN training efficiently in such a setting, several techniques have been proposed (Park et al., 2022; Waleffe et al., 2022). These studies assume that the graph and its features are stored on disks or SSDs and design their systems to reduce data transfers. The methods proposed in this paper directly apply to these settings by reducing bandwidth requirements, as seen in Section 4.

A single epoch of full-batch GNN training requires computation proportional to the number of layers $L$ and the size of the graph. However, minibatch training requires more operations to process a single epoch due to repeating calculations in the 2nd through $L$th layers. As the batch size decreases, the number of repeated calculations increases. This is because the vertices and edges have to be processed each time they appear in the $L$-hop neighborhood. Thus, it is natural to conclude that using effectively larger batch sizes in GNNs reduces the number of computations and data accesses of an epoch in contrast to regular DNN models. Our contributions in this work utilizing this important observation can be listed as follows:

- Investigate work vs. batch size relationship and present theorems stating the cost of processing a minibatch is a concave function of the batch size (Theorems 3.1 and 3.2).
- Utilize this relationship by combining data and intra-layer parallelism to process a minibatch across multiple PEs for reduced work (Section 3.1), with identical global batch size. We call this new approach *Cooperative Minibatching*.
- Use the same idea to generate consecutive dependent minibatches to increase temporal vertex embedding access locality (Section 3.2). This approach can reduce the transfer amount of vertex embeddings up to $4\times$, without harming model convergence.
- Show that the two approaches are orthogonal. Together, the reduced work and decreased cache miss rates result in up to $1.64\times$ speedup over Independent Minibatching with identical global batch size.

## 2 BACKGROUND

A graph $\mathcal{G} = (V, E)$ consists of vertices $V$ and edges $E \subset V \times V$ along with optional edge weights $A_{ts} > 0, \forall (t \to s) \in E$. Given a vertex $s$, the 1-hop neighborhood $N(s)$ is defined as $N(s) = \{t | (t \to s) \in E\}$, and it can be naturally expanded to a set of vertices $S$ as $N(S) = \cup_{s \in S} N(s)$.

GNN models work by passing previous layer embeddings ($H$) from $N(s)$ to $s$, and then combining them using a nonlinear function $f^{(l)}$ at layer $l$, given initial vertex features $H^{(0)}$:

$$H_s^{(l+1)} = f^{(l)}(H_s^{(l)}, \{H_t^{(l)} \mid t \in N(s)\}) \tag{1}$$

If the GNN model has $L$ layers, then the loss is computed by taking the final layer $L$'s embeddings and averaging their losses over the set of training vertices $V_t \subset V$ for *full-batch* training. In $L$-layer full-batch training, the total number of vertices that needs to be processed is $L|V|$.

### 2.1 MINIBATCHING IN GNNS

In minibatch training, a random subset of training vertices, called *Seed Vertices*, is selected, and training is done over the (sampled) subgraph composed of $L$-hop neighborhood of the seed vertices. On each iteration, minibatch training computes the loss on seed vertices, which are random subsets of the training set $V_t$.

Given a set of vertices $S$, we define $l$-th layer expansion set, or the $l$-hop neighborhood $S^l$ as:

$$S^0 = S, \quad S^{(l+1)} = S^l \cup N(S^l) \tag{2}$$

For GNN computations, $S^l$ would also denote the set of the required vertices to compute (1) at each layer $l$. Using the same notation, $\{s\}^l$ denotes $l$-layer expansion set starting from single vertex $s \in V$.

For a single minibatch iteration, the total number of vertices that need to be processed is $\sum_{l=1}^{L} |S^l|$. There are $\frac{|V|}{|S^0|}$ minibatches assuming $V_t = V$. Since each $|S^l| \geq |S^0|$, and a single epoch of minibatch training needs to go over the whole dataset, the work $W(|S^0|)$ for a single epoch is:

$$W(|S^0|) = \frac{|V|}{|S^0|} \sum_{l=1}^{L} E[|S^l|] \geq \frac{|V|}{|S^0|} \sum_{l=1}^{L} |S^0| = L|V| \tag{3}$$

where $E[|S^l|]$ is the expected number of sampled vertices in layer $l$ and $|S^0|$ is the batch size. That is, the total amount of work to process a single epoch increases over full-batch training. The increase in work due to minibatch training is thus encoded in the ratios $\frac{E[|S^l|]}{|S^0|}, 1 \leq l \leq L$.

Next, we will briefly present some of the sampling techniques. When sampling is used with minibatching, the minibatch subgraph may potentially become random. However, the same argument for the increasing total amount of work holds for them too, as seen in Figure 2.

## 2.2 GRAPH SAMPLING

Below, we review three different sampling algorithms for minibatch training of GNNs. Our focus in this work is samplers whose expected number of sampled vertices is a function of the batch size. All these methods are applied recursively for GNN models with multiple layers.

### 2.2.1 NEIGHBOR SAMPLING (NS)

Given a fanout parameter $k$ and a batch of seed vertices $S^0$, NS by (Hamilton et al., 2017) samples the neighborhoods of vertices randomly. Given a batch of vertices $S^0$, a vertex $s \in S^0$ with degree $d_s = |N(s)|$, if $d_s \leq k$, NS uses the full neighborhood $N(s)$, otherwise it samples $k$ random neighbors for the vertex $s$.

### 2.2.2 LABOR SAMPLING

Given a fanout parameter $k$ and a batch of seed vertices $S^0$, LABOR-0 (Balın & Çatalyürek, 2023) samples the neighborhoods of vertices as follows. First, each vertex rolls a uniform random number $0 \leq r_t \leq 1$. Given batch of vertices $S^0$, a vertex $s \in S^0$ with degree $d_s = |N(s)|$, the edge $(t \to s)$ is sampled if $r_t \leq \frac{k}{d_s}$. Since different seed vertices $\in S^0$ end up using the same random variate $r_t$ for the same source vertex $t$, LABOR-0 samples fewer vertices than NS in expectation.

The LABOR-* algorithm is the importance sampling variant of LABOR-0 and samples an edge $(t \to s)$ if $r_t \leq c_s \pi_t$, where $\pi$ is importance sampling probabilities optimized to minimize the expected number of sampled vertices and $c_s$ is a normalization factor. LABOR-* samples fewer vertices than LABOR-0 in expectation.

Note that, choosing $k \geq \max_{s \in V} d_s$ corresponds to training with full neighborhoods for both NS and LABOR methods.

### 2.2.3 RANDOMWALK SAMPLING

Given a walk length $o$, a restart probability $p$, number of random walks $a$, a fanout $k$, and a batch of vertices $S^0$, a vertex $s \in S^0$, a *Random Walk* (Ying et al., 2018) starts from $s$ and each step picks a random neighbor $s'$ from $N(s)$. For the remaining $o - 1$ steps, the next neighbor is picked from $N(s')$ with probability $1 - p$, otherwise it is picked from $N(s)$. This process is repeated $a$ times for each seed vertex and lastly, the top $k$ visited vertices become the *neighbors* of $s$ for the current layer.

Notice that random walks correspond to weighted neighbor sampling from a graph with adjacency matrix $\tilde{A} = \sum_{i=1}^{o} A^i$, where the weights of $\tilde{A}$ depend on the parameters $a$, $p$ and $k$. Random walks give us the ability to sample from $\tilde{A}$ without actually forming $\tilde{A}$.

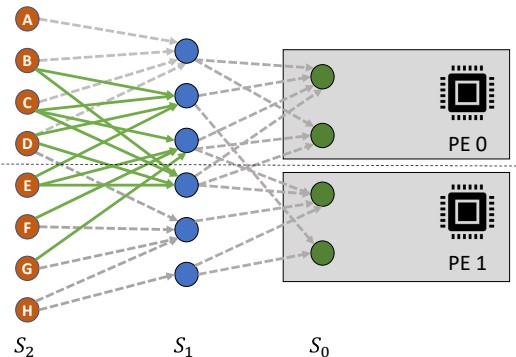

Figure 1: Minibatches of two processing elements may share edges in the second layer and vertices starting in the first layer. For independent minibatching, the solid green edges shared by both processing elements represent duplicate work, and input nodes B through G are duplicated along with the directed endpoints of green edges. However for cooperative minibatching, the vertices and edges are partitioned across the PEs with no duplication, and the edges crossing the line between the two PEs necessitate communication.

## 2.3 INDEPENDENT MINIBATCHING

Independent minibatching is commonly used in multi-GPU, and distributed GNN training frameworks (Cai et al., 2023; Gandhi & Iyer, 2021; Lin et al., 2020; Zheng et al., 2021; Zhu et al., 2019) to parallelize the training and allows scaling to larger problems. Each Processing Element (PE, e.g., GPUs, CPUs, or cores of multi-core CPU), starts with their own $S^0$ of size $b$ as the seed vertices, and compute $S^1, \ldots, S^L$ along with the sampled edges to generate minibatches (see Figure 1). Computing $S^1, \ldots, S^L$ depends on the chosen sampling algorithm, such as the ones explained in Section 2.2.

Independent minibatching has the advantage that doing a forward/backward pass does not involve any communication with other PEs after the initial minibatch preparation stage at the expense of duplicate work (see Figure 1).

## 3 COOPERATIVE MINIBATCHING

In this section, we present two theorems that show the work of an epoch will be monotonically nonincreasing with increasing batch sizes. We provide their proofs in Appendices A.1 and A.2. After that, we propose two algorithms that can take advantage of this monotonicity.

**Theorem 3.1.** *The work per epoch* $\frac{E[|S^l|]}{|S^0|}$ *required to train a GNN model using minibatch training is monotonically nonincreasing as the batch size* $|S^0|$ *increases.*

**Theorem 3.2.** *The expected subgraph size* $E[|S^l|]$ *required to train a GNN model using minibatch training is a concave function of batch size,* $|S^0|$.

### 3.1 COOPERATIVE MINIBATCHING

As explained in Section 2, Independent Minibatching (I.M.) can not take advantage of the reduction in work with increasing global batch sizes and number of PEs, because it uses separate small batches of sizes $b$ on each PE for each step of training. On the other hand, one can also keep the global batch size constant, $bP = |S^0|$, and vary the number of processors $P$. As $P$ increases, I.M. will perform more and more duplicate work because the local batch size is a decreasing function, $b = \frac{|S^0|}{P}$, of $P$.

Here, we propose the *Cooperative Minibatching* method that will take advantage of the work reduction with increasing batch sizes in multi-PE settings. In Cooperative Minibatching, a single batch of size $bP$ will be processed by all the $P$ PEs in parallel, eliminating any redundant work across all PEs.

We achieve this as follows: we first partition the graph in 1D fashion by logically assigning each vertex and its incoming edges to PEs as $V_p$ and $E_p$ for each PE $p$. Next, PE $p$ samples its batch seed vertices $S_p^l$ from the training vertices in $V_p$ for $l = 0$ of size $b$. Then using any sampling algorithm, PE $p$ samples the incoming edges $E_p^l$ from $E_p$ for its seed vertices. Each PE then computes the set of vertices sampled $\tilde{S}_p^{l+1} = \{t \mid (t \rightarrow s) \in E_p^l\}$. Note that, $\tilde{S}_p^{l+1}$ has elements residing on different PEs. The PEs exchange the vertex ids $\tilde{S}_p^{l+1}$ so that each PE receives the set $S_p^{l+1} \in V_p$. This process is repeated recursively for GNN models with multiple layers by using $S_p^{l+1}$ as the seed vertices for the next layer. The exchanged information is cached to be used during the forward/backward passes.

For the forward/backward passes, the same communication pattern used during cooperative sampling is used to send and receive input and intermediate layer embeddings before each GNN layer invocation. Algorithm 1 details cooperative sampling and cooperative forward/backward passes for a single GNN training iteration. Independent minibatching works the same except that it lacks the all-to-all operations and has $\tilde{A}_p^{l+1} = A_p^{l+1}$ for any given variable $A$ instead. The redistribution of vertices during sampling happens according to the initial graph partitioning and the rest of the redistribution operations follow the same communication pattern, always converting a variable $\tilde{A}_p^{l+1}$ into $A_p^{l+1}$ during the forward pass and $A_p^{l+1}$ into $\tilde{A}_p^{l+1}$ during sampling and the backward passes for any variable $A$. Note that a similar training approach is explored concurrently with our work in Polisetty et al. (2023). We refer the reader to Appendix A.4 for the complexity analysis of Cooperative and Independent Minibatching approaches, and to Appendix A.8 to see the relation between the approach proposed here and the work of Jia et al. (2020) on redundancy-free GCN aggregation.

## 3.2 Cooperative Dependent Minibatching

Just as any parallel algorithm can be executed sequentially, we can reduce the number of distinct data accesses by having a single PE process $b$-sized parts of a single $\kappa b$-sized minibatch for $\kappa$ iterations. In light of Theorems 3.1 and 3.2, consider doing the following: choose $\kappa \in \mathbb{Z}^+$, then sample a batch $S^0$ of size $\kappa b$, i.e., $\kappa b = |S^0|$ to get $S^0, \ldots, S^L$. Then sample $\kappa$ minibatches $S_i^0$, of size $b = |S_i^0|$ from this batch of size $\kappa b$ to get $S_i^0, \ldots, S_i^L, \forall i \in \{0, \ldots, \kappa - 1\}$. In the end, all of the input features required for these minibatches will be a subset of the input features of the large batch, i.e. $S_i^j \subset S^j, \forall i, j$. This means that the collective input feature requirement of these $\kappa$ batches will be $|S^L|$, the same as our batch of size $\kappa b$. Hence, we can now take advantage of the concave growth of the work in Theorem 3.2 and Figure 2.

Note that, if one does not use any sampling algorithms and proceeds to use the full neighborhoods, this technique will not give any benefits, as by definition, the $l$-hop neighborhood of a batch of size $\kappa b$ will always be equal to the union of the $l$-hop neighborhoods of batches of sizes $b$. However for sampling algorithms, any overlapping vertex sampled by any two batches of sizes $b$ might end up with different random neighborhoods resulting in a larger number of sampled vertices. Thus, having a single large batch ensures that only a single random set of neighbors is used for any vertex processed over a period of $\kappa$ batches.

The approach described above has a nested iteration structure and the minibatches part of one $\kappa$ group will be significantly different than another group and this might slightly affect convergence. In Appendix A.5, we propose an alternative smoothing approach that does not require two-level nesting and still takes advantage of the same phenomenon for the NS and LABOR sampling algorithms.

The main idea of our smoothing approach is as follows: each time one samples the neighborhood of a vertex, normally it is done independently of any previous sampling attempts. If one were to do it fully dependently, then one would end up with an identical sampled neighborhood at each sampling attempt. What we propose is to do something inbetween, so that the sampled neighborhood of a vertex changes slowly over time. The speed of change in the sampled neighborhoods is $\frac{1}{\kappa}$, and after every $\kappa$ iterations, one gets fully independent new random neighborhoods for all vertices. We will experimentally evaluate the locality benefits and the overall effect of this algorithm on convergence in Sections 4.2 and 4.3.1, and more details on our smoothing approach are discussed in Appendix A.5.

## 4 Experiments

We first compare how the work to process an epoch changes w.r.t. to the batch size to empirically validate Theorems 3.1 and 3.2 for different graph sampling algorithms. Next, we show how dependent batches introduced in Section 3.2 benefits GNN training. We also show the runtime benefits of cooperative minibatching compared to independent minibatching in the multi-GPU setting. Finally, we show that these two techniques are orthogonal, can be combined to get multiplicative savings. Details on our experimental setup can be found in Appendix A.3.

Table 1: Traits of datasets used in experiments: numbers of vertices, edges, avg. degree, features, cached vertex embeddings, and training, validation and test vertex split. The last column shows the number of minibatches in an epoch during model training with 1024 batch size including validation.

| Dataset | $|V|$ | $|E|$ | $\frac{|E|}{|V|}$ | # feats. | cache size | train - val - test (%) | # minibatches |
|---|---|---|---|---|---|---|---|
| flickr | 89.2K | 900K | 10.09 | 500 | 70k | 50.00 - 25.00 - 25.00 | 65 |
| yelp | 717K | 14.0M | 19.52 | 300 | 200k | 75.00 - 10.00 - 15.00 | 595 |
| reddit | 233K | 115M | 493.56 | 602 | 60k | 66.00 - 10.00 - 24.00 | 172 |
| papers100M | 111M | 3.2B | 29.10 | 128 | 2M | 1.09 -  0.11 -  0.19 | 1300 |
| mag240M | 244M | 3.44B | 14.16 | 768 | 2M | 0.45 -  0.06 -  0.04 | 1215 |

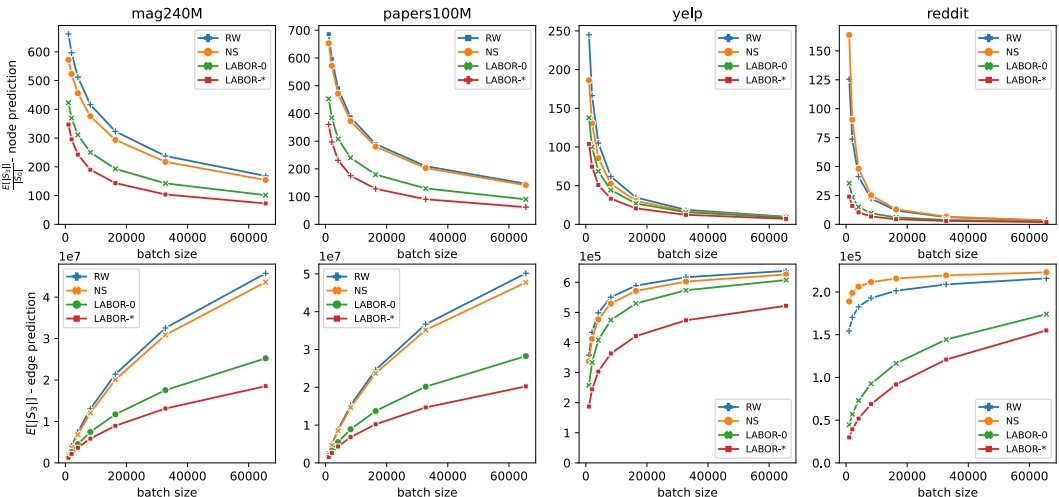

Figure 2: Monotonicity of the work. x-axis shows the batch size, y-axis shows $\frac{E[|S^3|]}{|S^0|}$ (see Theorem 3.1) for node prediction (top row) and $E[|S^3|]$ (see Theorem 3.2) for edge prediction (bottom row), where $E[|S^3|]$ denotes the expected number of sampled vertices in the 3rd layer and $|S^0|$ denotes the batch size. RW stands for Random Walks, NS for Neighbor Sampling, and LABOR-0/* for the two different variants of the LABOR sampling algorithm described in Section 2.2.

## 4.1 DEMONSTRATING MONOTONICITY OF WORK

We use three sampling approaches, NS, LABOR, and RW, to demonstrate that the work to process an epoch decreases as the batch size increases for the $L = 3$ layer case across these three different classes of sampling algorithms. We carried out our evaluation in two problem settings: node and edge prediction. For node prediction, a batch of training vertices is sampled with a given batch size. Then, the graph sampling algorithms described in Section 2.2 are applied to sample the neighborhood of this batch. The top row of Figure 2 shows how many input vertices is required on average to process an epoch, specifically $\frac{E[|S^3|]}{|S^0|}$. For edge prediction, we add reverse edges to the graph making it undirected and sample a batch of edges. For each of these edges a random negative edge (an edge that is not part of $E$) with one endpoint coinciding with the positive edge is sampled. Then, all of the endpoints of these positive and negative edges are used as seed vertices to sample their neighborhoods. The bottom row of Figure 2 shows $E[|S^3|]$.

We can see that in all use cases, datasets and sampling algorithms, the work to process an epoch is monotonically decreasing (see Appendix A.1 for the proof). We also see the plot of the expected number of vertices sampled, $E[|S^3|]$, is concave with respect to batch size (proof in Appendix A.2).

Another observation is that the concavity characteristic of $E[|S^3|]$ seems to differ for different sampling algorithms. In increasing order of concavity we have RW, NS, LABOR-0 and LABOR-*. The more concave a sampling algorithm's $E[|S^L|]$ curve is, the less it is affected from the NEP and more savings are available through the use of the proposed methods in Sections 3.1 and 3.2. Note that the differences would grow with a larger choice of layer count $L$.

## 4.2 DEPENDENT MINIBATCHES

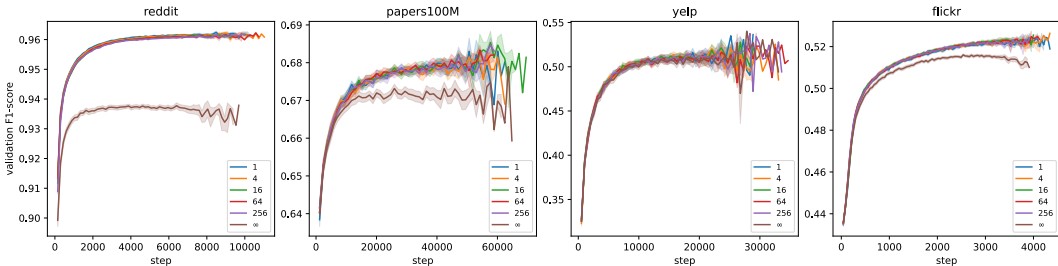

Figure 3: The validation F1-score with the full neighborhoods for LABOR-0 sampling algorithm with 1024 batch size and varying $\kappa$ dependent minibatches, $\kappa = \infty$ denotes infinite dependency, meaning the neighborhood sampled for a vertex stays static during training. See Figure 4a for cache miss rates. See Figure 7 for the validation F1-score with the dependent sampler and the training loss curve.

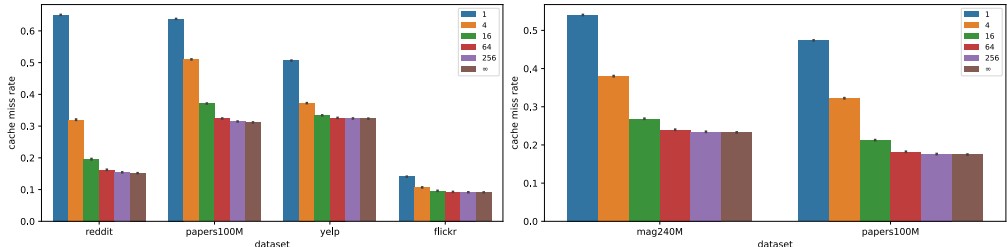

(a) Cache sizes were taken from Table 1 and a single PE was used.

(b) 4 cooperating PEs were used with each having a cache of size 1M.

Figure 4: LRU-cache miss rates for LABOR-0 sampling algorithm with 1024 batch size per PE and varying $\kappa$ dependent minibatches, $\kappa = \infty$ denotes infinite dependency.

We vary the batch dependency parameter $\kappa$ introduced in Section 3.2 for the LABOR-0 sampler with a batch size of 1024. Our expectation is that as consecutive batches become more dependent on each other, the subgraphs used during consecutive steps of training would start overlapping with each other, in which case, the vertex embedding accesses would become more localized. We attempted to capture this increase in temporal locality in vertex embedding accesses by implementing an LRU cache to fetch them, the cache sizes used for different datasets is given in Table 1. Note that the cache miss rate is proportional to the amount of data that needs to be copied from the vertex embedding storage. The Figure 4a shows that as $\kappa$ increases, the cache miss rate across all datasets drops. On reddit, this is a drop from 64% to 16% on, a 4x improvement. We also observe that the improvement is monotonically increasing as a function of $\frac{|E|}{|V|}$ given in Table 1. Figure 3 shows that training is not negatively affected across all datasets up to $\kappa = 256$ with less than 0.1% F1-score difference, after which point the validation F1-score with w/o sampling starts to diverge from the $\kappa = 1$ case. Runtime benefits of this approach can be observed by comparing the **Cache** and **Cache, $\kappa$** columns in Table 2. Appendix A.6 has additional discussion about the effect of varying $\kappa$ and the last column of Table 1 shows the number of minibatches in an epoch during training.

## 4.3 COOPERATIVE MINIBATCHING

We use our largest datasets, mag240M and papers100M, as distributed training is motivated by large-scale graph datasets. We present our runtime results on systems equipped with NVIDIA GPUs, with 4 and 8 A100 80 GB (NVIDIA, 2021) and 16 V100 32GB (NVIDIA, 2020b), all with NVLink interconnect between the GPUs (600 GB/s for A100 and 300 GB/s for V100). The GPUs perform all stages of GNN training and the CPUs are only used to launch kernels for the GPUs. Feature copies are performed by GPUs as well, accessing pinned feature tensors over the PCI-e using zero-copy access. In cooperative minibatching, both data size and computational cost are shrinking with increasing numbers of PEs, relative to independent minibatching. We use the GCN model for papers100M and the R-GCN model (Schlichtkrull et al., 2017) for mag240M. As seen in Table 2,

Table 2: Cooperative vs independent minibatching runtimes per minibatch (ms) on three different systems with 4 and 8 NVIDIA A100 80 GB GPUs, and 16 NVIDIA V100 32GB GPUs. I/C denotes whether independent or cooperative minibatching is used. Samp. is short for Graph Sampling, Feature Copy stands for vertex embedding copies over PCI-e and Cache denotes the runtime of copies performed with a cache that can hold $10^6$ vertex embeddings per A100 and $5 \times 10^5$ per V100. $\kappa$ denotes the use of batch dependency $\kappa = 256$. F/B means forward/backward. Total time is computed by the fastest available Feature Copy time, the sampling time, and the F/B time. $|S^0|$ is the global batch size and $b$ is the the batch size per GPU. $\alpha$ stands for cross GPU communication bandwidth (NVLink), $\beta$ for PCI-e bandwidth and $\gamma$ for GPU global memory bandwidth. Green was used to indicate the better result between independent and cooperative minibatching, while **Bold** was used to highlight the feature copy time included in the **Total** column.

| # PEs, $\gamma$ $\alpha, \beta, \|S^0\|$ | Dataset & Model | Sampler | I/C | Samp. | Feature Copy | | | F/B | Total |
|---|---|---|---|---|---|---|---|---|---|
| | | | | | - | Cache | Cache, $\kappa$ | | |
| 4 A100 $\gamma = 2\text{TB/s}$ $\alpha = 600\text{GB/s}$ $\beta = 64\text{GB/s}$ $\|S^0\| = 2^{12}$ $b = 1024$ | papers100M GCN | LABOR-0 | Indep | 21.7 | 18.4 | 16.8 | **11.2** | 8.9 | 41.8 |
| | | | Coop | 17.7 | 14.0 | 10.1 | **5.8** | 13.0 | **36.5** |
| | | NS | Indep | 16.1 | 26.5 | **22.1** | | 10.1 | 48.3 |
| | | | Coop | 11.9 | 21.3 | **12.9** | | 15.0 | **39.8** |
| | mag240M R-GCN | LABOR-0 | Indep | 26.0 | 57.9 | 56.0 | **41.0** | 199.9 | 266.9 |
| | | | Coop | 20.0 | 51.1 | 36.9 | **23.4** | 183.3 | **226.7** |
| | | NS | Indep | 14.4 | 78.0 | **71.2** | - | 223.0 | 308.6 |
| | | | Coop | 12.3 | 73.9 | **47.5** | - | 215.6 | **275.4** |
| 8 A100 $\gamma = 2\text{TB/s}$ $\alpha = 600\text{GB/s}$ $\beta = 64\text{GB/s}$ $\|S^0\| = 2^{13}$ $b = 1024$ | papers100M GCN | LABOR-0 | Indep | 21.3 | 21.1 | 18.7 | **12.0** | 9.3 | 42.6 |
| | | | Coop | 16.5 | 12.4 | 7.1 | **4.0** | 13.5 | **34.0** |
| | | NS | Indep | 15.8 | 31.0 | **24.5** | - | 10.3 | 50.6 |
| | | | Coop | 12.5 | 19.4 | **9.0** | - | 15.6 | **37.1** |
| | mag240M R-GCN | LABOR-0 | Indep | 30.6 | 70.1 | 66.2 | **46.8** | 202.1 | 279.5 |
| | | | Coop | 21.6 | 50.6 | 29.0 | **19.3** | 172.2 | **213.1** |
| | | NS | Indep | 15.0 | 94.9 | **80.9** | - | 224.8 | 320.7 |
| | | | Coop | 14.9 | 71.6 | **39.6** | - | 209.0 | **263.5** |
| 16 V100 $\gamma = 0.9\text{TB/s}$ $\alpha = 300\text{GB/s}$ $\beta = 32\text{GB/s}$ $\|S^0\| = 2^{13}$ $b = 512$ | papers100M GCN | LABOR-0 | Indep | 39.1 | 44.5 | 40.2 | **29.4** | 15.1 | 83.6 |
| | | | Coop | 26.9 | 22.7 | 10.4 | **4.9** | 19.1 | **50.9** |
| | | NS | Indep | 18.0 | 61.3 | **52.0** | - | 16.2 | 86.2 |
| | | | Coop | 19.2 | 34.9 | **13.0** | - | 21.3 | **53.5** |
| | mag240M R-GCN | LABOR-0 | Indep | 50.8 | 128.8 | 121.3 | **96.2** | 156.1 | 303.1 |
| | | | Coop | 29.2 | 78.1 | 42.8 | **23.5** | 133.3 | **186.0** |
| | | NS | Indep | 19.3 | 167.3 | **152.6** | - | 170.9 | 342.8 |
| | | | Coop | 19.3 | 116.1 | **53.1** | - | 160.4 | **232.8** |

cooperative minibatching reduces all the runtimes for different stages of GNN training, except for the F/B (forward/backward) times on papers100M where the computational cost is not high enough to hide the overhead of communication.

Table 3: Runtime improvements of Cooperative Minibatching over Independent Minibatching compiled from the **Total** column of Table 2. This is a further improvement on top of the speedup independent minibatching already gets over the execution on a single GPU.

| Dataset & Model | Sampler | 4 GPUs | 8 GPUs | 16 GPUs |
|---|---|---|---|---|
| papers100M GCN | LABOR-0 | 15% | 25% | 64% |
| | NS | 21% | 36% | 61% |
| mag240M R-GCN | LABOR-0 | 18% | 31% | 63% |
| | NS | 12% | 22% | 47% |

If we take the numbers in the **Total** columns from Table 2, divide independent runtimes by the corresponding cooperative ones, then we get Table 3. We can see that the theoretical decrease in work results in increasing speedup numbers with the increasing number of PEs, due to Theorem A.1. We would like to point out that $\frac{E[|S^3|]}{|S^0|}$ curves in Figure 2 are responsible for these results. With $P$

PEs and $|S^0|$ global batch size, the work performed by independent minibatching vs cooperative minibatching can be compared by looking at $x = \frac{1}{P}|S^0|$ vs $x = |S^0|$ respectively.

We also ran experiments that show that graph partitioning using METIS (Karypis & Kumar, 1998) prior to the start of training can help the scenarios where communication overhead is significant. The forward-backward time goes from 13.0ms to 12.0ms on papers100M with LABOR-0 on 4 NVIDIA A100 GPUs with such partitioning due to reduced communication overhead using the same setup as Table 2.

Increasing the number of GPUs increases the advantage of cooperative minibatching compared to independent minibatching. The forward-backward time on mag240M with LABOR-0 is 200 (same as independent baseline), 194, 187 and 183 ms with 1, 2, 3 and 4 cooperating PEs, respectively measured on the NVIDIA DGX Station A100 machine. The decrease in runtime with increasingly cooperating PEs is due to the decrease in redundant work they have to perform. Even though the batch size per PE is constant, the runtime goes down similar to the plots in the top row of Figure 2, except that it follows $\frac{kE[|S^2|]}{|S^0|}$, which gives the average number of edges in the 3rd layer when a sampler with fanout $k$ is used.

Additionally, we demonstrate that there is no significant model convergence difference between independent vs cooperative minibatching in Appendix A.7.

### 4.3.1 COOPERATIVE-DEPENDENT MINIBATCHING

Table 4: Runtime improvements of Dependent Minibatching for Independent and Cooperative Minibatching methods compiled from the **Cache, $\kappa$** and **Cache** columns of Table 2 with LABOR-0. Making consecutive minibatches dependent increases temporal locality, hence reducing cache misses.

| Dataset & Model | I/C | 4 GPUs | 8 GPUs | 16 GPUs |
|---|---|---|---|---|
| papers100M | Indep | 50% | 57% | 37% |
| GCN | Coop | 74% | 78% | 112% |
| mag240M | Indep | 37% | 41% | 26% |
| R-GCN | Coop | 58% | 50% | 82% |

We use the same experimental setup as Section 4.3 but vary the $\kappa$ parameter to show that cooperative minibatching can be used with dependent batches (Figure 4b). We use a cache size of 1M per PE. Cooperative feature loading effectively increases the global cache size since each PE caches only the vertices assigned to them while independent feature loading can have duplicate entries across caches. For our largest dataset mag240M, on top of $1.4\times$ reduced work due to cooperative minibatching alone, the cache miss rates were reduced by more than $2\times$, making the total improvement $3\times$. Runtime results for $\kappa \in \{1, 256\}$ are presented in Table 2, the Feature Copy **Cache** and **Cache, $\kappa$** columns. Table 4 summarizes these results by dividing the runtimes in **Cache** by **Cache, $\kappa$** and reporting percentage improvements.

## 5 CONCLUSIONS

In this paper, we investigated the difference between DNN and GNN minibatch training. We observed that the cost of processing a minibatch is a concave function of batch size in GNNs, unlike DNNs where the cost scales linearly. We then presented theorems that this is indeed the case for every graph and then proceeded to propose two approaches to take advantage of cost concavity. The first approach, which we call cooperative minibatching proposes to partition a minibatch between multiple PEs and process it cooperatively. This is in contrast to existing practice of having independent minibatches per PE, and avoids duplicate work that is a result of vertex and edge repetition across PEs. The second approach proposes the use of consecutive dependent minibatches, through which the temporal locality of vertex and edge accesses is manipulated. As batches get more dependent, the locality increases. We demonstrate this increase in locality by employing an LRU-cache for vertex embeddings on GPUs. Finally, we show that these approaches can be combined without affecting convergence, and speed up multi-GPU GNN training by up to $64\%$ for free.

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

# A APPENDIX

## A.1 WORK MONOTONICITY THEOREM

**Theorem A.1.** *The work per epoch required to train a GNN model using minibatch training is monotonically nonincreasing as the batch size increases.*

*Proof.* Given any $n \geq 2$, let's say we uniform randomly sample without replacement $S^0 \subset V$, where $n = |S^0|$. Now note that for any $S'^0 \subset S^0$, using the definition in (2), we have $S'^l \subset S^l, \forall l$. We will take advantage of that and define $S'^0 = S^0 \setminus \{s\}$ in following expression.

$$
\begin{aligned}
\sum_{\substack{s \in S^0 \\ S'^0 = S^0 \setminus \{s\}}} |S^l| - |S'^l| &= \sum_{\substack{s \in S^0 \\ S'^0 = S^0 \setminus \{s\}}} \sum_{w \in S^l} \mathbb{1}[w \notin S'^l] \\
&= \sum_{\substack{s \in S^0 \\ S'^0 = S^0 \setminus \{s\}}} \sum_{w \in \{s\}^l} \mathbb{1}[w \notin S'^l] \\
&= \sum_{\substack{s \in S^0 \\ S'^0 = S^0 \setminus \{s\}}} \sum_{w \in \{s\}^l} \mathbb{1}[w \notin \{s'\}^l, \forall s' \in S'^0] \\
&= \sum_{\substack{w \in S^l}} \sum_{\substack{s \in S^0 \\ w \in \{s\}^l}} \mathbb{1}[w \notin \{s'\}^l, \forall s' \in S^0 \setminus \{s\}]
\end{aligned}
\tag{4}
$$

In the last expression, for a given $w \in S^l$, if there are two different elements $s, s' \in S^0$ such that $w \in \{s\}^l$ and $w \in \{s'\}^l$, then the indicator expression will be 0. It will be 1 only if $w \in \{s\}^l$ for a unique $s \in S^0$. So:

$$
\begin{aligned}
\sum_{\substack{w \in S^l}} \sum_{\substack{s \in S^0 \\ w \in \{s\}^l}} \mathbb{1}[w \notin \{s'\}^l, \forall s' \in S^0 \setminus \{s\}] &= \sum_{\substack{w \in S^l \\ \exists! s \in S^0, w \in \{s\}^l}} 1 \\
&= |\{w \in S^l \mid w \in \{s\}^l, \exists! s \in S^0\}| \leq |S^l|
\end{aligned}
\tag{5}
$$

Using this, we can get:

$$
\begin{aligned}
\sum_{\substack{S'^0 \subset S^0 \\ |S'^0| + 1 = |S^0|}} |S^l| - |S'^l| &\leq |S^l| \\
\iff |S^0||S^l| - \sum_{\substack{S'^0 \subset S^0 \\ |S'^0| + 1 = |S^0|}} |S'^l| &\leq |S^l| \\
\iff |S^l|(|S^0| - 1) &\leq \sum_{\substack{S'^0 \subset S^0 \\ |S'^0| + 1 = |S^0|}} |S'^l| \\
\iff |S^l|(|S^0| - 1) &\leq |S^0| E[|S'^l|] \\
\iff \frac{|S^l|}{|S^0|} &\leq \frac{E[S'^l]}{|S'^0|}
\end{aligned}
$$

Since $S^0$ was uniformly randomly sampled from $V$, its potential subsets $S'^0$ are also uniformly randomly picked from $V$ as a result. Then, taking an expectation for the random sampling of $S^0 \subset V$, we conclude that $\frac{E[|S^l|]}{|S^0|} \leq \frac{E[|S'^l|]}{|S'^0|}$, i.e., the expected work of batch size $n$ is not greater than the

work of batch of size $n - 1$. This implies that the work with respect to batch size is a monotonically nonincreasing function. □

Empirical evidence can be seen in Figures 2 and 5. In fact, the decrease is related to the cardinality of the following set:

$$T_l(S) = \{w \in S^l \mid w \in \{s\}^l, \exists! s \in S^0\}$$

When $T(S^0)$ is equal to $S^l$, the work is equal as well. In the next section, we further investigate the effect of $|T(S^0)|$ on $E[|S^l|]$.

## A.2 OVERLAP MONOTONICITY

In addition to the definition of $T(S)$ above, if we define the following set $T^2(S)$:

$$T_2^l(S) = \{w \in S^l \mid w \in \{s\}^l \cap \{s'\}^l, \exists! \{s, s'\} \subset S^0\}$$

**Theorem A.2.** *The expected subgraph size $E[|S^l|]$ required to train a GNN model using minibatch training is a concave function of batch size, $|S^0|$.*

*Proof.* Given any $n \geq 2$, let's say we uniformly randomly sample without replacement $S^0 \subset V$ of size $n$.

$$
\begin{aligned}
|T_l(S^0)| - 2|T_2^l(S^0)| &= \sum_{\substack{S'^0 \subset S^0 \\ |S'^0|+1=|S^0|}} |T_l(S^0)| - |T_l(S'^0)| \\
&= |S^0||T_l(S^0)| - |S^0|E[|T_l(S'^0)|] \\
\Longleftrightarrow (|S^0| - 1)|T_l(S^0)| &= |S^0|E[|T_l(S'^0)|] - 2|T_2^l(S^0)| \\
\Longleftrightarrow \frac{|T_l(S^0)|}{|S^0|} &= \frac{E[|T_l(S'^0)|]}{|S'^0|} - \frac{2|T_2^l(S^0)|}{|S'^0||S^0|} \\
\Longrightarrow \frac{|T_l(S^0)|}{|S^0|} &\leq \frac{E[|T_l(S'^0)|]}{|S'^0|}
\end{aligned}
\tag{6}
$$

where the first equality above is derived similar to Equations (4) and (5). Overall, this means that the overlap between vertices increases as the batch size increases. Utilizing our finding from Equations (4) and (5), we have:

$$
\begin{aligned}
\sum_{\substack{S'^0 \subset S^0 \\ |S'^0|+1=|S^0|}} |S^l| - |S'^l| &= |T_l(S^0)| \\
\Longrightarrow |S^0||S^l| - |S^0|E[|S'^l|] &= |T_l(S^0)| \\
\Longrightarrow |S^l| &= E[|S'^l|] + \frac{|T_l(S^0)|}{|S^0|} \\
\Longrightarrow E[|S^l|] &= E[|S'^l|] + \frac{E[|T_l(S^0)|]}{|S^0|}
\end{aligned}
\tag{7}
$$

Note that the last step involved taking expectations for the random sampling of $S^0$. See the recursion embedded in the equation above, the expected size of the subgraph $S^l$ with batch size $|S^0|$ depends on the expected size of the subgraph $S'^l$ with batch size $|S^0| - 1$. Expanding the recursion, we get:

$$E[|S^l|] = \sum_{i=1}^{|S^0|} \frac{E[|T_l(V_0^i)|]}{i} \tag{8}$$

where $V_0^i$ is a random subset of $V$ of size $i$. Since $\frac{E[|T_l(V_0^i)|]}{i}$ is monotonically nonincreasing as $i$ increases as we showed in (6), we conclude that $E[|S^l|]$ is a concave function of the batch size, $|S^0|$. $\qquad\square$

So, the slope of the expected number of sampled vertices flattens as batch size increases, see the last row in Figure 2 and the first row in Figure 5. Note that this implies work monotonicity as well.

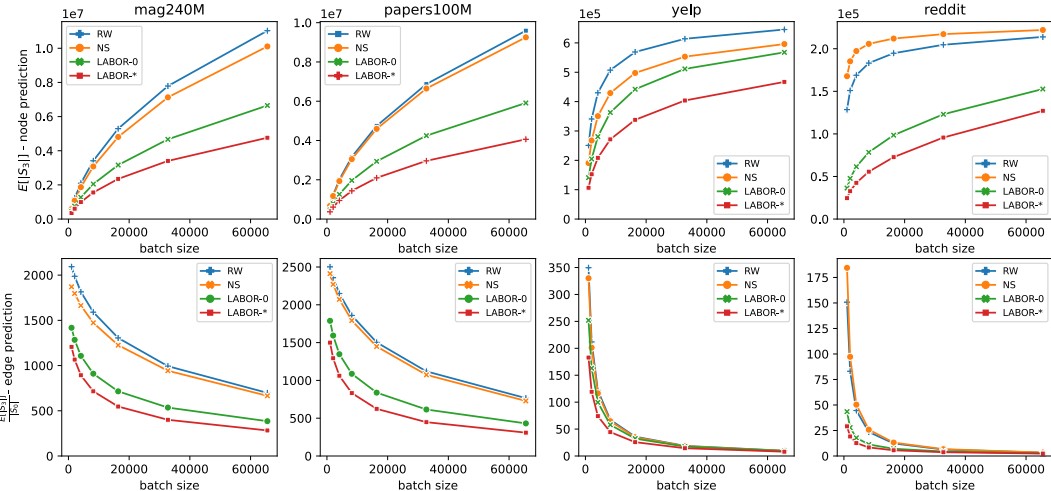

Figure 5: Monotonicity of the work. x axis shows the batch size, y axis shows $E[|S^3|]$ for node prediction (top row) and $\frac{E[|S^3|]}{|S^0|}$ for edge prediction (bottom row), where $E[|S^3|]$ denotes the expected number of vertices sampled in the 3rd layer and $|S^0|$ denotes the batch size. RW stands for Random Walks, NS stands for Neighbor Sampling, and LABOR-0/* stand for the two different variants of the LABOR sampling algorithm described in Section 2.2. Completes Figure 2.

### A.3 Experimental Setup

**Setup:** In our experiments, we use the following datasets: reddit (Hamilton et al., 2017), papers100M (Hu et al., 2020), mag240M (Hu et al., 2021), yelp and flickr (Zeng et al., 2020), and their details are given in Table 1. We use Neighbor Sampling (NS) (Hamilton et al., 2017), LABOR Sampling (Balın & Çatalyürek, 2023) and Random Walks (RW) (Ying et al., 2018) to form minibatches. We used a fanout of $k = 10$ for the samplers. In addition, Random Walks used length of $o = 3$, restart probability $p = 0.5$ and number of random walks from each seed $a = 100$. All our experiments involve a GCN model with $L = 3$ layers (Hamilton et al., 2017), with 1024 hidden dimension for mag240M and papers100M and 256 for the rest. Additionally, papers100M and mag240M datasets were made undirected graphs for all experiments and this is reflected in the reported edge counts in Table 1. Input features of mag240M are stored with the 16-bit floating point type. We use the Adam optimizer (Kingma & Ba, 2014) with $10^{-3}$ learning rate in all the experiments.

**Implementation:** We implemented[1] our experimental code using C++ and Python in the DGL framework (Wang et al., 2019) with the Pytorch backend (Paszke et al., 2019). All our experiments were repeated 50 times and averages are presented. Early stopping was used during model training runs. So as we go to the right along the x-axis, the variance of our convergence plots increases because the number of runs that were ongoing is decreasing.

### A.4 Complexity Analysis

Let $M(V_1, E, V_2)$ denote the work to process a bipartite graph $V_2 \rightarrow V_1$ with edges $E$ for a given GNN model $M$. Assuming cross PE communication bandwidth $\alpha$, Storage (e.g., disk, network, or DRAM) to PE bandwidth as $\beta$ and PE memory bandwidth $\gamma$, and cache miss rate $\rho$, we have the time

---

[1]Source code is available in the supplementary material.

Table 5: Algorithmic complexities of different stages of GNN training at layer $0 \leq l < L$ with $L$ total layers and batch size $B = |S^0|$ with $P$ PEs. Note that $|S_p^l(B)| = |S^l(B)|\frac{1}{P}$, $|E_p^l(B)| = |E^l(B)|\frac{1}{P}$ since the PEs process the partitioned subgraphs. Feature loading happens only at layer $L$.

| Stage | Independent | Cooperative |
|---|---|---|
| Sampling | $\mathcal{O}(|S^l(\frac{B}{P})|\frac{1}{\beta})$ | $\mathcal{O}(|S_p^l(B)|\frac{1}{\beta} + |\tilde{S}_p^{l+1}(B)|\frac{c}{\alpha})$ |
| Feature loading | $\mathcal{O}(|S^L(\frac{B}{P})|\frac{d\rho}{\beta})$ | $\mathcal{O}(|S_p^L(B)|\frac{d\rho}{\beta} + |\tilde{S}_p^L(B)|\frac{dc}{\alpha})$ |
| Forward/Backward | $\mathcal{O}(M(S^l(\frac{B}{P}), E^l(\frac{B}{P}), S^{l+1}(\frac{B}{P}))\frac{d}{\gamma})$ | $\mathcal{O}(M(S_p^l(B), E_p^l(B), \tilde{S}_p^{l+1}(B))\frac{d}{\gamma} + |\tilde{S}_p^{l+1}(B)|\frac{dc}{\alpha})$ |

complexities given in Table 5 to perform different stages of GNN training per PE. We also use $d$ for embedding dimension and $c < 1$ for the cross edge ratio, note that $c \approx \frac{P-1}{P}$ for random partitioning, and smaller for smarter graph partitioning with $P$ standing for the number of PEs. Also the sizes of $\tilde{S}^l$ become smaller when graph partitioning is used due to increased overlap.

---

**Algorithm 1** Cooperative minibatching

---

**Input:** seed vertices $S_p^0$ for each PE $p \in P$, # layers $L$
**for all** $l \in \{0, \ldots, L-1\}$ **do** {Sampling}
  **for all** $p \in P$ **do in parallel**
    Sample next layer vertices $\tilde{S}_p^{l+1}$ and edges $E_p^l$ for $S_p^l$
    all-to-all to redistribute vertex ids for $\tilde{S}_p^{l+1}$ to get $S_p^{l+1}$
**for all** $p \in P$ **do in parallel** {Feature Loading}
  Load input features $H_p^L$ from Storage for vertices $S_p^L$
  all-to-all to redistribute $H_p^L$ to get $\tilde{H}_p^L$
**for all** $l \in \{L-1, \ldots, 0\}$ **do** {Forward Pass}
  **for all** $p \in P$ **do in parallel**
    **if** $l+1 < L$ **then**
      all-to-all to redistribute $H_p^{l+1}$ to get $\tilde{H}_p^{l+1}$
    Forward pass on bipartite graph $\tilde{S}_p^{l+1} \to S_p^l$ with edges $E_p^l$, input $\tilde{H}_p^{l+1}$ and output $H_p^l$
**for all** $p \in P$ **do in parallel**
  Compute the loss and initialize gradients $G_p^0$
**for all** $l \in \{0, \ldots, L-1\}$ **do** {Backward Pass}
  **for all** $p \in P$ **do in parallel**
    Backward pass on bipartite graph $S_p^l \to \tilde{S}_p^{(l+1)}$ with edges $E_p^l$, input $G_p^l$ and output $\tilde{G}_p^{l+1}$
    **if** $l+1 < L$ **then**
      all-to-all to redistribute $\tilde{G}_p^{l+1}$ to get $G_p^{l+1}$

---

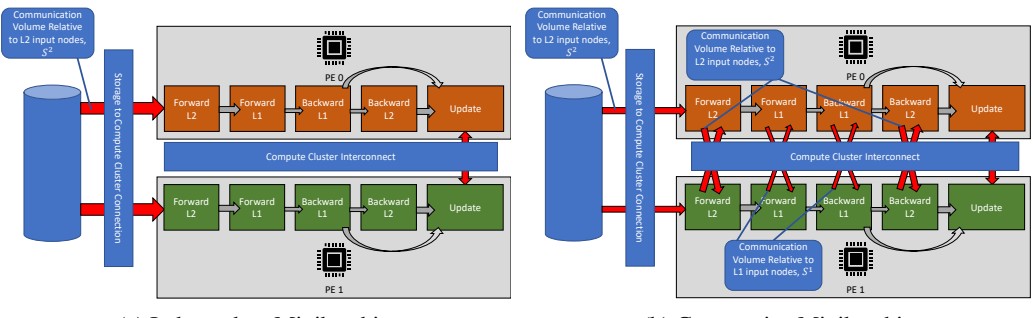

(a) Independent Minibatching          (b) Cooperative Minibatching

Figure 6: A comparison of Independent and Cooperative Minibatching approaches in the feature loading and forward-backward GNN stages. The thickness of the red arrows indicates the data volume. Due to redundant vertices across PEs, Independent Minibatching wastes (PCI-e) bandwidth for vertex embedding copies from Storage. Moreover, the PEs perform identical computations for redundant edges across PEs in Independent Minibatching.

Our goal in this section is to empirically show the work reduction enjoyed by cooperative minibatching over independent minibatching by reporting the number of vertices and edges processed per PE. We also report the number of vertices that are communicated for cooperative minibatching during its all-to-all calls in Algorithm 1 and Figure 6b. The results are given in Table 6.

Table 6: Average number of vertices and edges sampled in different layers with LABOR-0 per PE, reduced by taking the maximum over 4 PEs (All the numbers are in thousands, lower is better) with batch size $|S^0| = 1024$. $c|\tilde{S}^l|$ shows the number of vertices communicated at layer $l$. Papers and mag were used as short versions of papers100M/GCN and mag240M/R-GCN dataset model pairs respectively. Last column shows forward-backward (F/B) runtime in ms.

| Dataset | Part. | I/C | $|S^3|$ | $c|\tilde{S}^3|$ | $|\tilde{S}^3|$ | $|E^2|$ | $|S^2|$ | $c|\tilde{S}^2|$ | $|\tilde{S}^2|$ | $|E^1|$ | $|S^1|$ | F/B |
|---|---|---|---|---|---|---|---|---|---|---|---|---|
| | random | Indep | 463 | 0 | 463 | 730 | 74.8 | 0 | 74.8 | 93.6 | 9.63 | 8.9 |
| papers | random | Coop | 318 | 311 | 463 | 608 | 62.4 | 56.8 | 82.8 | 89.9 | 9.28 | 13.0 |
| | metis | Coop | 328 | 179 | 402 | 615 | 63.1 | 34.0 | 73.8 | 90.8 | 9.35 | 12.0 |
| | random | Indep | 443 | 0 | 443 | 647 | 67.9 | 0 | 67.9 | 82.0 | 8.78 | 199.9 |
| mag | random | Coop | 324 | 310 | 459 | 566 | 59.8 | 53.1 | 77.3 | 80.4 | 8.62 | 183.3 |
| | metis | Coop | 334 | 178 | 419 | 576 | 60.6 | 31.0 | 71.3 | 81.8 | 8.80 | 185.1 |

Looking at Tables 5 and 6, we make the following observations:

1. All runtime complexities for cooperative minibatching scales with $|S_p^l(B)| = |S^l(B)|\frac{1}{P}$ and $|E_p^l(B)| = |E^l(B)|\frac{1}{P} \leq |S^l(B)|\frac{k}{P}$ and for independent minibatching with $|S^l(\frac{B}{P})|$ and $|E^l(\frac{B}{P})| \leq |S^l(\frac{B}{P})|k$, for a sampler with fanout $k$. Since $E[|S^l(B)|]$ is a concave function, $E[|S^l(B)|]\frac{1}{P} \leq E[|S^l(\frac{B}{P})|]$, and this corresponds to looking at Figure 2 first row with $x = B$ for coop and $x = \frac{B}{P}$ for independent if one wanted to guess how their runtime would change with changing $B$ and $P$. For an example, all the runtime numbers we have provided in the Table 2 are for 4 GPUs. Going from 4 to 8 would increase the edge of cooperative over independent even more, see Table 3.

2. Sampling and Feature loading over PCI-e requires $\alpha \gg \beta$ for cooperative to get a speedup over independent.

3. In order for cooperative F/B to improve against independent, we need that $\frac{\alpha}{c} > \frac{\gamma}{M}$.

4. Cross edge ratio $c$ reduces all communication between PEs. In particular, graph partitioning will lower both $c$ and $|\tilde{S}_p^l(B)|$, lowering the communication overhead, see $c|\tilde{S}^l|$ columns in Table 6.

5. The model complexity $M$ is small for the GCN model (papers100M) but large for the R-GCN model (mag240M), as shown by the F/B runtime numbers in Table 2. Also, the communication overhead between the two scenarios is similar, meaning communication can take from upto 30% to less than a few percent depending on $M$. For the papers100M dataset, communication makes up more of the runtime, so graph partitioning helps bring the F/B runtime down. However, the load imbalance caused by graph partitioning slows down the F/B runtime despite lowered communication costs for the mag240M dataset.

In today's modern multi-GPU systems we see that $\gamma \approx 2$ TB/s , $\alpha \approx 300$ GB/s and $\beta \approx 30$ GB/s (NVIDIA, 2020a). Due to $\alpha$ being relatively fast compared to $\frac{\gamma}{M}$ and $\beta$, our approach is feasible. On newer systems, the all-to-all bandwidth continues to increase (NVIDIA, 2023), decreasing the cost of cooperation on a global mini-batch.

However, on systems where the interconnect does not provide full bandwidth for all-to-all operations, our approach is limited in the speedup it can provide. Our approach is most applicable for systems with relatively fast alltoall bandwidth $\frac{\alpha}{c}$ compared to $\frac{\gamma}{M}$ and $\beta$ and large $P$. In particular, starting from $P = 2$, cooperative starts to outperform independent even on F/B with the mag240M dataset and the R-GCN model in Section 4.3 and Tables 2 and 3.

### A.5 Smoothed Dependent Minibatching

As described in Section 2, NS algorithm works by using the random variate $r_{ts}$ for each edge $(t \to s)$. Being part of the same minibatch means that a single random variate $r_{ts}$ will be used for each edge. To generate these random variates, we initialize a Pseudo Random Number Generator (PRNG) with a random seed $z$ along with $t$ and $s$ to ensure that the first rolled random number $r_{ts}$ from the PRNG stays fixed when the random seed $z$ is fixed. Given random seeds $z_1$ and $z_2$, let's say we wanted to use $z_1$ for the first $\kappa$ iterations and would later switch to the seed $z_2$. This switch can be made smoothly by interpolating between the random seeds $z_1$ and $z_2$ while ensuring that the resulting sampled random numbers are uniformly distributed. If we are processing the batch number $i < \kappa$ in the group of $\kappa$ batches, then we want the contribution of $z_2$ to be $c = \frac{i}{\kappa}$, while the contribution of $z_1$ is $1 - c$. We can sample $n_{ts}^1 \sim \mathcal{N}(0, 1)$ with seed $z_1$ and $n_{ts}^2 \sim \mathcal{N}(0, 1)$ with seed $z_2$. Then we combine them as

$$n_{ts}(c) = \cos(\frac{c\pi}{2})n_{ts}^1 + \sin(\frac{c\pi}{2})n_{ts}^2$$

note that $n_{ts}(0) = n_{ts}^1$, $n_{ts}(1) = n_{ts}^2$ and $n_{ts}(c) \sim \mathcal{N}(0, 1), \forall c \in \mathbb{R}$ also, then we can set $r_{ts} = \Phi(n_{ts}(c))$, where $\Phi(x) = \mathbb{P}(Z \le x)$ for $Z \sim \mathcal{N}(0, 1)$, to get $r_{ts} \sim U(0, 1)$ that the NS algorithm can use. Dropping $s$ from all the notation above gives the version for LABOR. In this way, the random variates change slowly as we are switching from one group of $\kappa$ batches to another. When $i = \kappa$, we let $z_1 \leftarrow z_2$ and initialize $z_2$ with another random seed. To use this approach, only the random variate generation logic needs modification, making its implementation for any sampling algorithm straightforward compared to the nested approach initially described.

### A.6 Dependent batches (cont.)

Looking at the training loss and validation F1-score with sampling curves in Figure 7, we notice that the performance gets better as $\kappa$ is increased. This is due to the fact that a vertex's neighborhood changes slower and slower as $\kappa$ is increased, the limiting case being $\kappa = \infty$, in which case the sampled neighborhoods are unchanging. This makes training easier so $\kappa = \infty$ case leads the pack in the training loss and validation F1- score with sampling curves.

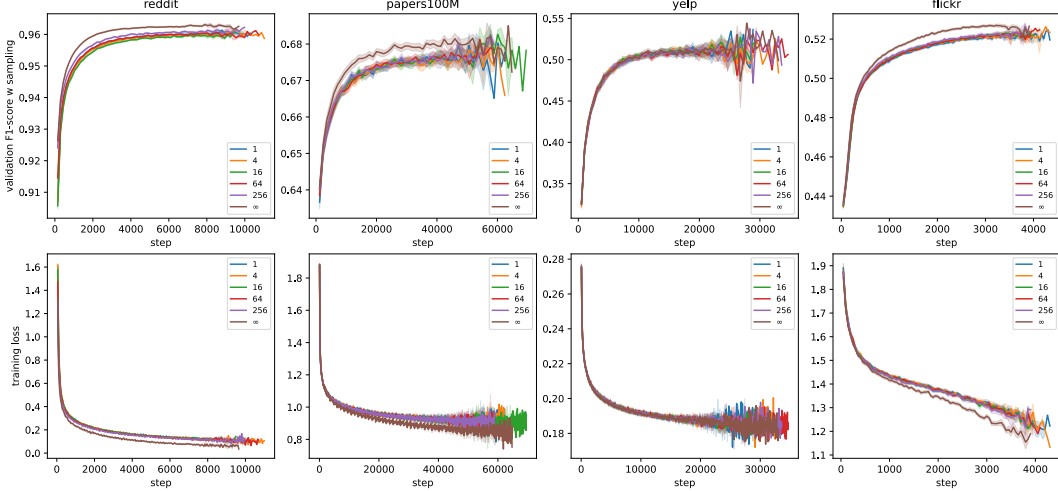

Figure 7: LABOR-0 sampling algorithm with 1024 batch size and varying $\kappa$ dependent minibatches, $\kappa = \infty$ denotes infinite dependency, meaning the neighborhood sampled for a vertex stays static during training. The first row shows the validation F1-score with the dependent sampler. The second row shows the training loss curve. Completes Figure 3.

### A.7 Comparing a single batch vs $P$ independent batches convergence

We investigate whether training with a single large batch in $P$-GPU training shows any convergence differences to the current approach of using $P$ separate batches for each of the GPUs. We use a

global batch size of $4096$ and divide a batch into $P \leq 8$ independent batches, with each batch having a size of $\frac{4096}{P}$. We use NS and LABOR-0 samplers with fanouts of $k = 10$ for each of the 3 layers. Figure 8 shows that there are no significant differences between the two approaches, we present the results averaged over the samplers to save space.

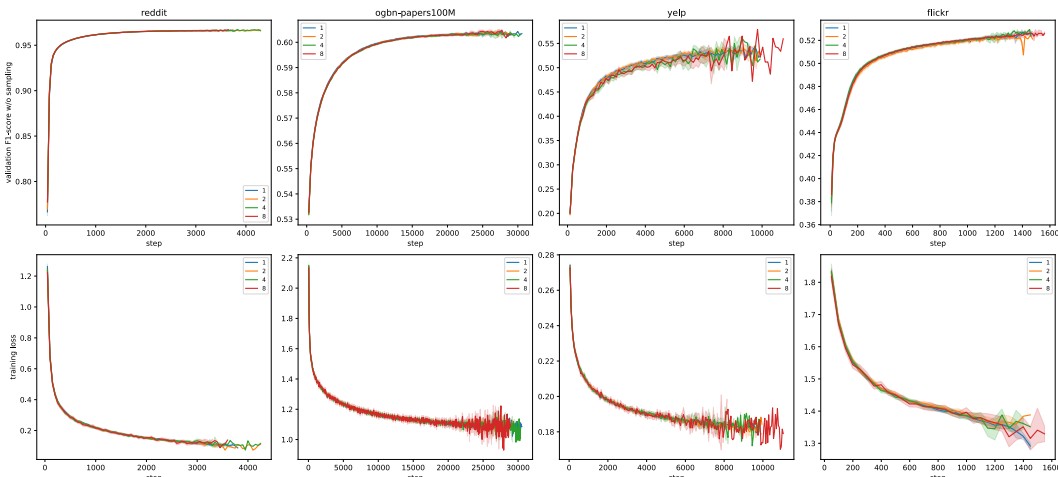

Figure 8: Convergence difference between cooperative vs independent minibatching with a global batch size of 4096 averaged over Neighbor and LABOR-0 samplers.

## A.8 REDUNDANCY FREE GCN AGGREGATION

Jia et al. (2020) proposes a method of reducing processing during neighborhood aggregation by finding common sub-neighborhoods among aggregated vertices, whether using full-batch or minibatch training. That is, if two vertices have the neighborhoods $\{A, B, C\}$ and $\{B, C, D\}$, and a summation operator is used for aggregation, then instead of computing four additions: $A + B + C$ and $B + C + D$ concurrently, the three additions $BC = B + C$, $A + BC$, and $BC + D$ can be computed. This approach is orthogonal to the approaches proposed in Section 3 in that it reduces redundant aggregation steps, whereas our approach reduces redundant input nodes and edges in parallel computations. As such, the two approaches could be employed together.

## A.9 SUMMARY OF THE MAIN INSIGHTS OF COOPERATIVE MINIBATCHING

In Section 2, the work $W(|S^0|)$ to process an epoch (full pass over the dataset) for a given minibatch size $|S^0|$ is characterized by the number of minibatches in an epoch ($\frac{|V|}{|S^0|}$) $\times$ the amount of work to process a single minibatch, which is approximated by the sum of the number of sampled vertices in each layer ($\sum_{l=1}^{L} |S^l|$). This can be seen in Equation (3).

Equation (3) only considers the number of processed vertices and it is good enough for our purposes. Since all the sampling algorithms we consider in Section 2.2 have fanout parameters $k$, the number of edges sampled for each seed vertex has an upper bound $k$. So, given vertices $S^l$ for the $l$th layer, the number of sampled edges in that same layer will be $\leq k|S^l|$. Clearly for almost any GNN model, the runtime complexity to process layer $l$ is linearly increasing w.r.t. the number of vertices ($|S^l|$) and edges ($\leq k|S^l|$) processed, so the runtime complexity is $\mathcal{O}(|S^l| + k|S^l|) = \mathcal{O}(|S^l|)$.

A more comprehensive analysis of the runtime complexities of Cooperative and Independent Minibatching approaches is provided in Appendix A.4, taking into account the exact numbers of sampled vertices ($|S^l|$), edges ($|E^l|$), and various communication bandwidths ($\alpha, \gamma, \beta$) and even graph partition quality $c$ and cache misses $\rho$.

As Cooperative Minibatching considers a single minibatch of size $B$ for all $P$ PEs, the growth of the number of sampled vertices and edges is characterized by $B$ as the minibatch size. In contrast, Independent Minibatching assigns different minibatches of sizes $\frac{B}{P}$ to each PE, so the growth of the sampled vertices and edges is characterized by $\frac{B}{P}$ as the minibatch size. Due to Theorems 3.1 and 3.2,

the work $W(B)$ with respect to a minibatch size $B$ is a concave function, we have $W(B) \leq PW(\frac{B}{P})$, meaning that Cooperative Minibatching is theoretically faster than Independent Minibatching, if communication between PEs is infinitely fast. For empirical data, one can look at the first-row of Figure 2 at the $x$-axis $B$ for Cooperative and $\frac{B}{P}$ for Independent, due to:

$$W(B) \leq PW(\frac{B}{P}) \iff \frac{W(B)}{B} \leq \frac{W(\frac{B}{P})}{\frac{B}{P}}$$

as proven by Theorem 3.1. As $\frac{E[|S^3|]}{|S^0|}$ curves in Figure 2 are decreasing, the work of Cooperative Minibatching is significantly less than Independent Minibatching. Finally, modern multi-GPU computer systems have very fast inter-GPU communication bandwidths, e.g. NVLink, which is why we are able to show favorable results compared to Independent Minibatching despite performing seemingly unnecessary communication.

