# OpenReview forum: "Cooperative Minibatching in Graph Neural Networks"
_ICLR.cc/2024/Conference — Submitted to ICLR 2024_

### Official Review · Reviewer_LvyT · 2023-10-29

**Soundness:** 1 poor
**Presentation:** 1 poor
**Contribution:** 1 poor
**Rating:** 1
**Confidence:** 5

**Summary:**

In this paper, the authors propose a cooperative mini-batching design that utilizes the overlap of sampled subgraphs. The propose approach claims to optimize the communication during GNN training.

**Strengths:**

Optimizing the communication during GNN training is an important topic.

**Weaknesses:**

—The writing is informal and not well organized, which is reflected but not limited to the following aspects: 1) ambiguous terminology and inconsistent word choice, e.g., "work" (not clear what this really means); 2) overly extensive background (not sure how most content in the background is related to the target problem); 3) causal usage of theorem.

—The theoretical result states that the expected work increases as batch size increases, which clearly depends on the setting. For example, what if, in the extreme case,  the communication speed is infinite? What if I am using different GNN models? The theoretical result does not provide any specification with this regard and it is not clear how one can prove such a result.  In addition, it is not clear how the theoretical results motivate or connect with the proposed approach.

—No discussion on related works, despite there exists an extensive line of research from both algorithm and system communities that try to optimize distributed/large-scale GNN training. For example, I found the idea is closely related to layer sampling that is commonly used in GNN, and the communication optimization proposed in [1].

[1] Gandhi, Swapnil, and Anand Padmanabha Iyer. "P3: Distributed deep graph learning at scale." 15th USENIX Symposium on Operating Systems Design and Implementation (OSDI), 2021.

**Questions:**

The theoretical result states that the expected work increases as batch size increases, which clearly depends on the setting. For example, what if, in the extreme case,  the communication speed is infinite? What if I am using different GNN models? The theoretical result does not provide any specification with this regard and it is not clear how one can prove such a result.  In addition, it is not clear how the theoretical results motivate or connect with the proposed approach.

---

> ### Author Response · Authors · 2023-11-15
>
> Thank you very much for your feedback. Let us clarify some of the weaknesses pointed out and reply to your questions below.
>
> Weakness 1: We will work on increasing the formality of our paper to make it read better. We will also consider moving some of the background into the appendix to make more space for other content in the final revision.
> To reply to your question, the “work” refers to the “computation" that needs to be carried out for sequential execution, as it is usually referred to in computational complexity.
> Hence, in this context, for a given minibatch size, it is a quantity that monotonically increases as the number of sampled vertices and edges increases, and the details of this dependency are characterized by the underlying GNN model.
> Appendix A.4 and Table 5 go more into detail about what we mean by work.
> However, simply looking at how many vertices and edges will be sampled for different batch sizes is a good enough approximation to reason about the final runtime for any GNN model.
> We know that there will be redundant edges for independent minibatching meaning it will be slower in theory.
>
>
> Weakness 2 and Questions: Appendix A.4 along with Table 5 characterizes the work more formally. There, we take into account the GNN model complexity and the different bandwidths available in a
> computer system (PCI-e, inter-GPU bandwidth, GPU memory bandwidth).
> To reply to your question, if communication speed (inter-GPU bandwidth) is infinite, then cooperative minibatching will be guaranteed to be faster than independent minibatching. Given $P$ PEs, the
> main complexity difference comes from the fact that Independent Minibatching has $\frac{B}{P}$ as the local batch size for each PE but Cooperative Minibatching has $B$ as the
> global batch size, which PEs cooperate to process together (note that the global batch size is $B$ for both methods).
> If $W$ denotes the work for a given minibatch size, we know that $P W(\frac{B}{P}) \geq W(B)$ due to the theorems
> we prove in our paper. The paragraph right after Table 3 in our paper already makes the connection between the empirical curves in Figure 2 and the observed runtime
> results in Tables 2 and 3. We will make the necessary modifications to our paper to make this more clear.
>
>
> Furthermore, our proposed approaches are agnostic to the underlying GNN model, as only the layer outputs in between the layers need to be communicated when using Cooperative Minibatching,
> because every GNN model has vertex embeddings as the input to the layer and output of the layer. The GCN and R-GCN models we use are the computationally least expensive GNN models
> for homogenous and heterogenous graphs. If we were to use GAT or R-GAT models in our experimental evaluation, because these models are computationally slower, the communication runtime
> would be even less significant increasing the advantage of Cooperative Minibatching compared to Independent.
>
>
> Weakness 3: Note that we already cite [1] in our paper. There, they propose to fetch different channels of vertex features from other PEs (We always partition across vertices),
> while duplicating the mini-batch graph structure on each processing element for the very first layer (or last layer with the sampling-focused notation in our paper), performing GNN aggregation operation in an intra-layer model parallelism fashion for only a single layer.
>
>
> Note that, the approach used in [1] is not as easily generalizable to different GNN models such as Graph Attention Networks, which they point out require special handling.
> On the contrary, our proposed Cooperative Minibatching approach is agnostic to the GNN model and requires no special handling.
> In the same first layer, [1] switches to data parallelism (Independent Minibatching) as can be seen in Figure 5 in [1].
> In our work, we propose to process all stages of GNN training including graph sampling, feature loading, and all layers of forward/backward
> stages in a cooperative manner as the redundancy argument applies at all stages of GNN training. Moreover, it is possible to even combine the intra-layer parallelism approach in
> [1] with our proposed approach.
>
>
> [1] Gandhi, Swapnil, and Anand Padmanabha Iyer. "P3: Distributed deep graph learning at scale." 15th USENIX Symposium on Operating Systems Design and Implementation (OSDI), 2021.

---

### Official Review · Reviewer_m5Sh · 2023-10-31

**Soundness:** 3 good
**Presentation:** 3 good
**Contribution:** 3 good
**Rating:** 6
**Confidence:** 3

**Summary:**

This paper addresses the issue of redundant data access and computation across processing elements (PEs) in the context of independent minibatch Graph Neural Network (GNN) training. The authors conduct both empirical and theoretical investigation for the monotonicity of work size and the concavity of the expected subgraph size needed for GNN training, with respect to the batch size. Based on these two properties, the author introduces cooperative and dependent mini-batching methodologies. These strategies are designed to minimize computational redundancy and optimize temporal locality for data access. The evaluations on several datasets with different system setups show that the proposed techniques can speedup multi-GPU GNN training by up to 64%.

**Strengths:**

+ The paper is well-written and addresses a practical issue in minibatch GNN training.
+ The proposed methodologies are strongly motivated from both empirical and theoretical perspectives.
+ The paper demonstrates significant performance improvement with minimal overhead. The appendix serves as a good supplement to the main content.

**Weaknesses:**

- The paper lacks sufficient evaluation to demonstrate the generalizabiliy and scalability of the proposed technique.
- The paper omits some relevant citations for related work that should be discussed and compared.
- There are some points need further clarifications.

**Questions:**

1. The paper only evaluates the proposed method on one GNN model. How generalizable is this method to other types of GNNs, especially those deep GNNs with tens of layers?
2. The parameter (\kappa) for batch dependency is set as 256, however, Figure 4 indicates minimal difference between 64, 256, and even infinity. Furthermore, Figure 3 shows that the GNN model validation F1-score drops when \kappa is 256 (or larger). Given these observations, how do you justify the choice of \kappa as 256 for the evaluation instead of 64?
3. Could you clarify the unit of measurement for the cache size? Is it quantified in bytes or in terms of the number of vertices/features? What are the key factors for determining the optimal cache size for different datasets, GNN models, and the hardware platforms?
4. What are the communication cost with and without applying the proposed technique?
5. The work in [1] also solves the redundancy issue across the PEs using cooperative training. However, this related work is not cited or compared in this paper.
6. It might be beneficial to include more illustrative figures, especially, for the algorithm 1. This would help readers to follow the steps of the proposed method more easily.

[1]. GSplit: Scaling Graph Neural Network Training on Large Graphs via Split-Parallelism

---

> ### Author Response · Authors · 2023-11-15
>
> Thank you very much for your feedback. Let us reply to your questions below:
>
> 1. In our experiments, we used GCN and R-GCN, which are the simplest and fastest GNN models for homograph datasets (a single edge type), and heterographs (multiple edge types).
> Our proposed methods are agnostic to the underlying GNN model. The more complicated and higher complexity the GNN model is, the more benefit we will see from
> Cooperative Minibatching as the reduction in work due to redundant computations will outweigh the cost of the communication even further. Moreover, the more layers there are, the more overlap
> there will be between multiple PEs, meaning that Cooperative Minibatching will be even more beneficial. We chose 3 layers as a realistic number that people use in industrial
> deployments of GNNs. Our proposed methods will apply to any GNNs that utilize minibatch training
> with graph sampling (Cooperative and Dependent) or with full neighborhoods (Cooperative only).
>
>
> 2. As you have already pointed out, Figure 4b shows that the cache miss rate difference between 64 and 256 is almost indistinguishable.
> So, we could have very well chosen $\kappa=64$, and our runtime results would have ended up being the same.
> The lower $\kappa$ is, the less the chance that it will negatively affect convergence. We agree that using $\kappa=64$ is a good option as well.
> A good rule of thumb would be to use a $\kappa$ value that gives almost the same cache miss rate as $\kappa=\infty$, and the smallest such $\kappa$ should be chosen.
>
>
> 3. Table 1 reports the cache sizes, and its unit is in the number of vertices whose features are cached. Thus, the size of the cache would be cache-size times \#feats times 4 bytes
> for the float32 datatype. The optimal cache size is the largest size you can pick without running out of memory on your system and PEs. We ran our code in different scenarios with
> different-sized datasets. For example, we could have cached all of the Reddit dataset but then we would not have been able to demonstrate the benefits of dependent minibatching on that dataset
> as the cache miss rate would have been 0\% for all $\kappa$ values.
>
>
> 4. The communication costs are provided in Appendix A.4. In particular, Table 5 gives the complexities of different GNN stages taking into account the PCI-e bandwidth, the
> NVLink bandwidth and the GPU memory bandwidth along with the number of sampled edges and vertices in each layer while also taking into account the underlying GNN model computational complexity.
>
>
> 5. Thank you for pointing us to a concurrent work that we were not aware of. Indeed the split parallelism and cooperative training method proposed in [1] are very similar to our Cooperative Minibatching method.
> The differences we see are that while we propose cooperative minibatching for all stages of GNN training (sampling, feature loading, and forward/backward),
> [1] proposes it only for feature loading and forward/backward stages. Furthermore, they are not proving the theorems we prove in our paper and they are not proposing the
> dependent minibatching approach. We have cited this work as concurrent work in our rebuttal revision. However, the fact that similar approaches are showing up in the literature
> provides further proof that our proposed methods are credible and useful.
>
> We started this work in June 2022, the first version of our code has been publicly available since August 2022 and our first submission of this paper was in January 2023,
> which we can prove after the double-blind review period.
>
>
> 6. We are working on more illustrative figures for the camera-ready revision of our paper.
>
>
> [1]. GSplit: Scaling Graph Neural Network Training on Large Graphs via Split-Parallelism

---

> ### Author Response · Authors · 2023-11-19
> **New figure added**
>
> We added Figure 6 to Appendix A.4 to illustrate the difference between Independent and Cooperative Minibatching methods and accompany algorithm 1 as per your suggestion. We will continue to work on reorganizing our paper for the camera-ready version to see if we can swap some of the more relevant content from the Appendix with content from the main paper according to your and the other reviewers' suggestions.

---

### Official Review · Reviewer_GNH8 · 2023-11-04

**Soundness:** 2 fair
**Presentation:** 3 good
**Contribution:** 2 fair
**Rating:** 6
**Confidence:** 3

**Summary:**

The paper identifies an issue with standard mini-batching approaches for training graph neural networks: that redundant computations are performed by processors due to them sharing edges and vertices. A new method, cooperative mini-batching, is proposed, in which processors jointly process a single global mini-batch, exchanging information as needed to avoid redundant computation. Independent versus cooperative mini-batching is then compared experimentally.

**Strengths:**

1. The paper identifies an important difference between mini-batching in standard DNN training and in GNNs, which I had not before observed, and proposes a method to avoid inefficiencies in independent mini-batch training in GNNs. This has the potential to positively impact the training of GNNs.
2. There is both theoretical and experimental justification for this method.
3. The paper includes extensive experiments to support its points.

**Weaknesses:**

1. The paper seems to be missing a discussion of the communication costs of its method, which seem like they could be significant, especially in a multi-node setup.. This is in contrast to independent mini-batching, which has the nice property of avoiding communication. The appendix (A3) notes communication overhead is not an issue, but this could be measured in detail, and in any case only considers a small-scale, on-node scenario.
2. How is the 1D partitioning done, in detail? Appendix A3 notes that using METIS was beneficial, so why not always use this (or a similar graph partitioning algorithm)?
3. The paper does not discuss the memory overhead of cooperative mini-batching. It seems to me that all samples in a K-hop neighborhood of each vertex need to be present for the method, which seems like it would result in significant memory overheads, especially as the number of layers increases (the paper only considers a network with three layers).
4. The paper seems to be only considering neighborhood sampling as a way of performing independent mini-batching. However, there are other ways to do this, e.g., graph cut algorithms. (See, for example, Rozemberczki et al., "Little Ball of Fur: A Python Library for Graph Sampling", CIKM 2020.)

**Questions:**

1. What are the communication costs of cooperative mini-batching? How does the method perform in cross-node scenarios?
2. How is partitioning done? Why not always use a method that reduces cross-device edges?
3. How does cooperative mini-batching scale with the depth of the network?

-----

I thank the authors for their clarifications, and have slightly raised my score accordingly. I would, however, echo the concerns of reviewer LvyT regarding the text.

---

> ### Author Response · Authors · 2023-11-15
>
> Thank you very much for your feedback on our paper. Let us first address some of the weaknesses you pointed out and then provide answers to your questions.
>
> Weakness 1 and Question 1: Appendix A.4 and Table 5 go into more detail about the complexities of Independent vs Cooperative Minibatching.
> There, the complexity is characterized by the number of sampled edges and vertices in each layer (tied directly to the theorems and empirical figures in our paper).
> We also talk about when cooperative minibatching is a better choice than independent minibatching taking into account the available bandwidths of the computer system.
>
> But to summarize, the important thing is the GNN model's computational complexity and relative speed of
> communication in the computer system. In our experiments, we use two different
> GNN models, GCN, and R-GCN, however, our approach works with any GNN model as it only requires routing the layer outputs
> in between each layer. GCN and R-GCN are computationally
> faster models compared to for example GAT and R-GAT models (Graph Attention Networks) due to their use of the
> computationally more expensive SDDMM kernels (GCN uses SPMM only). Thus, our choice of GCN and R-GCN models is
> the worst-case scenario for Cooperative Minibatching. In our experimental evaluation, we selected single-node
> multi-GPU systems as these systems have NVLink providing relatively fast communication, the bandwidths are
> provided in Table 2, $\gamma$ and $\alpha$ give us the memory bandwidth of the GPU and the communication bandwidth
> across GPUs respectively, which are used in Appendix A.4 to characterize the complexities of communication and computation
> of each GNN training stage.
>
> New computer systems have increasingly more efficient communication. The new hardware on the market by NVIDIA
> provides NVLink connections that work in the multi-node setting for up to 256 GPUs [1] providing intra-node-like bandwidth
> across nodes. However, we do not have access to such systems. As our method strongly depends on the existence of
> a high bandwidth communication between PEs, it will not bring benefits in any arbitrary setting. In our case, NVLink bandwidth
> is $\alpha=600$GB/s vs $\gamma=2$TB/s of the GPU global memory bandwidth for the 8 GPU DGX-A100 system in Table 2.
> Note that multi-GPU systems by AMD and Intel have equivalent high-bandwidth intra-node GPU interconnects.
>
> Weakness 2 and Question 2: We randomly permute the vertices of the original graph, then logically assign equal-sized
> contiguous ranges of vertices to each PE in our main experiments. Even with such naive partitioning, we show
> favorable runtime results. The problem with graph partitioning is that it is hard to ensure load balance for the GNN setting.
> Even if you partition the graph while load-balancing the number of vertices or edges in each partition, GNN
> computation specifically requires the L-hop neighborhoods of training vertices to be load-balanced.
>
> Our results in Table 6 show the load
> balance issues with METIS partitioning, even though the runtime on the papers dataset decreases from 13ms to 12ms, we also see that the
> runtime goes from 183ms to 185ms on the mag dataset with partitioning. This is due to the fact that on papers, the faster GCN model is used, where
> the communication overhead is relatively higher, so reducing communication improves runtime. However, for the R-GCN model used for the mag dataset,
> we see a slowdown, because the computation runtime is dominant, and ensuring load balance takes priority over reducing communication.
> We note that our experimentation with graph partitioning is
> preliminary and follow-up research is required to ensure that Cooperative Minibatching can be made even faster
> than it already is, which we will leave as future work. Our goal in this paper is to show that Cooperative Minibatching
> is a viable alternative to Independent Minibatching and to motivate further research on it.
>
> Weakness 3 and Question 3: The only memory overhead of cooperative minibatching compared to independent minibatching is the allocated
> buffers for the all-to-all communication calls. Otherwise, because there are no duplicate vertices or edges
> on the PEs in Cooperative Minibatching, Independent Minibatching has higher memory overhead, which is
> exacerbated when the number of GPUs is increased. Moreover, as the number of layers is increased, the overlap
> between the 4-hop neighborhoods will be even greater than the overlap for 3-hop neighborhoods. Thus, if there were 4 layers,
> the 4-layer equivalent of Table 2 would show even more favorable results for Cooperative Minibatching. We do not readily
> have access to the computer systems we used in our experiments so we can not easily provide you with the empirical verification.
>
>
> [1] https://www.nvidia.com/en-us/data-center/nvlink/

---

> > ### Author Response · Authors · 2023-11-15
> >
> > Weakness 4: We have tried to incorporate the well-established GNN sampling methods available in the literature.
> > NS is the first work proposing sampling to train GNNs. Random walks were also proposed to sample from l-hop (l > 1)
> > neighborhood instead of the immediate 1-hop neighborhood as in NS. Finally, LABOR is a very recent GNN
> > sampling method that is similar but ensures more overlapping vertices compared to NS. Sampling in the general
> > graph setting may be richer, however, there are complications for GNNs as one usually wants the sampling
> > method to approximate training without sampling in an unbiased manner. Thus, parts of the richer
> > graph-sampling literature may not be applicable to GNNs and our goal is not to explore new ways to perform sampling for GNNs.

---

### Author Response · Authors · 2023-11-15

We thank all the reviewers for their valuable feedback. We would like to note that in our paper, we are comparing two distinct methodologies of GNN training,
Independent and our proposed Cooperative Minibatching. We started with the Independent Minibatching code available in DGL and modified it to use Cooperative Minibatching
for all stages of GNN training to make the comparison fair without any extra optimizations.


In this work, we want to prove that Cooperative Minibatching is a choice that GNN system implementers need to consider to get the maximum performance from a single-node
multi-GPU system, or any similar system where the communication speed is fast relative to the rest of the system. In the multi-node setting, where inter-node communication is slower
than intra-node communication, one can utilize the Cooperative Minibatching method inside the nodes and the Independent Minibatching method across multiple nodes.


We randomly permute the vertices in the graph and logically assign contiguous chunks of vertices to different PEs, ensuring perfect load balance but high communication. We also have a preliminary evaluation of METIS partitioning,
however, it is hard to ensure load balancing. Even if you partition the graph while load-balancing the number of vertices or edges in each partition, GNN
computation specifically requires the L-hop neighborhoods of training vertices to be load-balanced, which requires investigating GNN-specialized partitioning methods.
The use of METIS showed practical benefits for the papers100M/GCN scenario where the GNN model is computationally inexpensive, so reducing communication
showed runtime improvements. However, it led to slowdowns for the mag240M/R-GCN scenario due to R-GCN being expensive so load balancing takes priority over reducing communication.
These results can be found in Table 6 in Appendix section A.4. Thus, our main experimental results are with random partitioning. Despite the use of random partitioning, we already see
benefits from Cooperative Minibatching in the multi-GPU setting which we wanted to share with the larger GNN community.


In our rebuttal revision, we have moved some of the experimental details to the appendix to make space for the preliminary METIS partitioning discussion.
Also, we have added a new Table 4 to highlight the benefits of our proposed dependent minibatching approach to summarize Table 2. We also added Figure 6 to Appendix A.4 to illustrate the difference between Independent and Cooperative Minibatching methods.
Due to the limited amount of time, we do not have the chance to make large changes to our paper to address all your reviews during the rebuttal period.


For the camera-ready version of our paper, our plan is to improve the organization and terminology of our paper as pointed out by reviewer LvyT,
and incorporate the clarifications we made to the reviewers in the main paper, and add further discussion in the appendix if we are space-limited.

---

### Author Response · Authors · 2023-11-23
**Clarification about work and computational complexity to reviewers LvyT and GNH8**

Dear reviewers,

We want to clarify the work definition (computational complexity) in our manuscript. In Section 2, the work $W(|S^0|)$ to process an epoch (full pass over the
dataset) for a given minibatch size $|S^0|$ is characterized by the number of minibatches
in an epoch ($\frac{|V|}{|S^0|}$) $\times$ the amount of work to process a single
minibatch, which is approximated by the sum of the number of sampled vertices in
each layer ($\sum_{l=1}^L |S^l|$). This can be seen in Equation (3).

Equation (3) only considers the number of processed vertices and it is good
enough for our purposes. Since all the sampling algorithms we consider
in Section 2.2 have fanout parameters $k$, the number of edges sampled
for each seed vertex has an upper bound $k$. So, given vertices $S^l$ for the $l$th layer,
the number of sampled edges in that same layer will be $\leq k |S^l|$. Clearly for almost
any GNN model, the runtime complexity to process layer $l$ is linearly increasing w.r.t.
the number of vertices ($|S^l|$) and edges $(\leq k|S^l|)$ processed, so the computational
complexity becomes:

$\mathcal{O}(|S^l| + k|S^l|) = \mathcal{O}(|S^l|)$

A more comprehensive analysis of the runtime complexities of Cooperative and
Independent Minibatching approaches is provided in Appendix A.4,
taking into account the exact numbers of sampled vertices ($|S^l|$), edges ($|E^l|$),
and various communication bandwidths ($\alpha$, $\gamma$, $\beta$) and even graph
partition quality $c$ and cache misses $\rho$.
Appendix A.4 is formal and detailed enough to reason whether Cooperative Minibatching would outperform Independent Minibatching
on any given computer system and GNN training scenario.

As Cooperative Minibatching considers a single minibatch of size $B$ for all $P$ PEs, the
growth of the number of sampled vertices and edges is characterized by $B$ as the
minibatch size. In contrast, Independent Minibatching assigns different minibatches of
sizes $\frac{B}{P}$ to each PE, so the growth of the sampled vertices and edges is
characterized by $\frac{B}{P}$ as the minibatch size. Due
to Theorems 3.1 and 3.2, the work $W(B)$ with respect to a
minibatch size $B$ is a concave function, we have $W(B) \leq PW(\frac{B}{P})$, meaning
that Cooperative Minibatching is theoretically faster than Independent Minibatching if
communication between PEs is infinitely fast. For empirical data, one can look at the
first-row of Figure 2 at the $x$-axis $B$ for Cooperative and $\frac{B}{P}$ for Independent, due to:

$W(B) \leq PW(\frac{B}{P}) \iff \frac{W(B)}{B} \leq \frac{W(\frac{B}{P})}{\frac{B}{P}}$

as proven by Theorem 3.1. As $\frac{E[|S^3|]}{|S^0|}$ curves
in Figure 2 are decreasing, the work of Cooperative Minibatching is
significantly less than Independent Minibatching. Finally, modern multi-GPU computer
systems have very fast inter-GPU communication bandwidths, e.g. NVLink, which is why we
are able to show favorable results compared to Independent Minibatching despite performing
seemingly unnecessary communication.

We included the discussion above in Appendix A.9 as a "Summary of the main insights of Cooperative Minibatching".
Parts of the discussion above can be found or inferred from the main paper alone. Due to space and time limitations,
we are not able to move this section to the main paper as is.

---

### Meta-Review · Area_Chair_VJoG · 2023-12-14

**Metareview:**

The paper proposes a cooperative mini-batching scheme for GNN training, where the main idea is to allow processors collaborate on the same minibatch to improve training efficiency. While the idea is interesting and the paper has been evaluated on several large-scale datasets, the reviewers and AC found it lacking in discussion of existing scalable GNN training work. They felt a major revision would be necessary to better align and compare the proposed method with established approaches. Therefore, we recommend rejection for this paper.

Detailed comments: Reviewer m5Sh and LvyT both pointed out the absence of discussion regarding several relevant existing works. Moreover, it remains unclear whether the authors are familiar with the widely used standard graph sampling approach for scalable GNN training. Approaches like GraphSaint and Cluster-GCN propose forming subgraphs at each iteration and restricting propagation within them, significantly reducing memory and computational costs. GraphAutoScale further explores scaling to massive graphs with out-of-memory features. This scheme is also prevalent in various GNN training libraries. The paper lacks clarity on how the proposed cooperative mini-batching scheme could be integrated with this established approach.

**Justification For Why Not Higher Score:**

Lack of discussions/comparisons to previous work.

**Justification For Why Not Lower Score:**

N/A

---

### Decision · Program_Chairs · 2024-01-16

Reject